# Fungal Proteins from *Sargassum* spp. Using Solid-State Fermentation as a Green Bioprocess Strategy

**DOI:** 10.3390/molecules27123887

**Published:** 2022-06-17

**Authors:** Adriana M. Bonilla Loaiza, Rosa M. Rodríguez-Jasso, Ruth Belmares, Claudia M. López-Badillo, Rafael G. Araújo, Cristóbal N. Aguilar, Mónica L. Chávez, Miguel A. Aguilar, Héctor A. Ruiz

**Affiliations:** 1Biorefinery Group, Food Research Department, School of Chemistry, Autonomous University of Coahuila, Saltillo C.P. 25280, Coahuila, Mexico; adriana_bonilla@uadec.edu.mx (A.M.B.L.); ruthbelmares@uadec.edu.mx (R.B.); cllopezb@uadec.edu.mx (C.M.L.-B.); rafael.araujo@uadec.edu.mx (R.G.A.); cristobal.aguilar@uadec.edu.mx (C.N.A.); monicachavez@uadec.edu.mx (M.L.C.); 2Centro de Investigación y de Estudios Avanzados del Instituto Politécnico Nacional, Unidad Saltillo, Av. Industria Metalúrgica 1062, Ramos Arizpe C.P. 25900, Coahuila, Mexico; miguel.aguilar@cinvestav.edu.mx

**Keywords:** seaweed, macroalgal biomass, bioreactor, hydrothermal pretreatment, bioprocess

## Abstract

The development of green technologies and bioprocesses such as solid-state fermentation (SSF) is important for the processing of macroalgae biomass and to reduce the negative effect of *Sargassum* spp. on marine ecosystems, as well as the production of compounds with high added value such as fungal proteins. In the present study, *Sargassum* spp. biomass was subjected to hydrothermal pretreatments at different operating temperatures (150, 170, and 190 °C) and pressures (3.75, 6.91, and 11.54 bar) for 50 min, obtaining a glucan-rich substrate (17.99, 23.86, and 25.38 g/100 g d.w., respectively). The results indicate that *Sargassum* pretreated at a pretreatment temperature of 170 °C was suitable for fungal growth. SSF was performed in packed-bed bioreactors, obtaining the highest protein content at 96 h (6.6%) and the lowest content at 72 h (4.6%). In contrast, it was observed that the production of fungal proteins is related to the concentration of sugars. Furthermore, fermentation results in a reduction in antinutritional elements, such as heavy metals (As, Cd, Pb, Hg, and Sn), and there is a decrease in ash content during fermentation kinetics. Finally, this work shows that *Aspergillus oryzae* can assimilate nutrients found in the pretreated *Sargassum* spp. to produce fungal proteins as a strategy for the food industry.

## 1. Introduction

Currently, the coasts of the Caribbean Sea are invaded by *Sargassum* spp, including *Sargassum natans*, *Sargassum fluitans*, and *Sargassum muticum*. These blooms have occurred specifically in the Caribbean Sea, the western Central Atlantic, and the Gulf of Mexico [1]. One of the causes of the blooms is the size and frequency of seaweed worldwide [2,3]. The large amounts of *Sargassum* present in the Caribbean and the Gulf of Mexico produce negative economic, social, and environmental impacts. Among the impacts is the decomposition of seaweed in coastal areas, producing hydrogen sulfide and ammonia that can result in complications for human health [4,5]. Moreover, *Sargassum* provides habitat for more than 127 endangered species, with the Sargasso Sea absorbing approximately 7% of global net carbon emissions per year [6]. *Sargassum* and brown macroalgae are rich in minerals, water-soluble polysaccharides, and phenolic compounds that improve soil, health, quality, productivity, and enzyme activities [7,8,9]. Therefore, *Sargassum* biomass can be used in the production of biochemicals, biofuels, and pharmaceutical products in terms of a biorefinery and circular bioeconomy [10,11]. Likewise, for the food industry, alginates from *Sargassum* are extracted and added to food as gelling, thickening, encapsulating, and coating agents [12]. Farhan et al. [13] studied the probiotic potential of *Sargassum polycystum* as a supplement in diets, finding favorable results in prebiotic function when the diets of fingerlings in Asia were supplemented at a concentration that does not exceed 1.5 and 3.0 g/100 g of feed [13]. On the other hand, the processing and fractionation of this macroalgal biomass are important to develop feasible processes. Hydrothermal pretreatment or processing is a technology that uses high pressures to fractionate biomass, applying different operating temperatures and times for hydrolysis, extraction of compounds, and chemical and structural modification of macroalgal biomass [14]. Hydrothermal pretreatments for biomass fractionation are important and promising technologies, and these can be scaled up at the pilot and industrial levels under different forms of operation (batch and continuous) [15,16,17]. In addition, the integration of bioprocesses such as SSF is necessary for the recovery and conversion of macroalgal biomass into high-added-value compounds [18]. SSF is a bioprocess that has become relevant in recent years, as it produces beneficial transformations or conversion of biomass using biochemical platforms using *Trichoderma* and *Aspergillus fungi* as substrates [18,19,20]. Thus, filamentous fungi are used in solid-state fermentation, such as *Rhizopus oryzae* considered Generally Recognized as Safe (GRAS) by the USDA in the production of organic acids, enzymes, volatile compounds, antioxidant compounds and fungal proteins [21,22,23]. The objective of this study was the biotransformation of *Sargassum* through hydrothermal processes and solid-state fermentation to obtain fungal proteins from *Aspergillus* for food industry application.

## 2. Results

### 2.1. Physicochemical and Bromatological Characterization of the Biomass of Sargassum spp.

The chemical composition (g/100 g *Sargassum* (d.w.)) of the dry biomass of *Sargassum* spp. was glucan (11.64 ± 0.01), galactan (0.99 ± 0.01), fucoidan (3.18 ± 0.01), protein (4.3 ± 0.07), insoluble acid residue (29 ± 0.01), ash (15.46 ± 0.40) and moisture (13.22 ± 0.20) (Table 1). In other words, *Sargassum*, due to its biochemical composition, can be used as a substrate for the metabolism of microorganisms through fermentation, and hydrothermal pretreatments, to produce value-added compounds for health, agriculture, and food.

Variability of the structural polysaccharides of *Sargassum* spp. was observed, with the highest concentration of glucan, followed by fucoidan and galactan. Glucan (as laminarin), galactan, and fucoidan were analyzed.

### 2.2. Characterization of the Elemental Composition of Sargassum spp.

The elemental composition [magnesium (Mg), phosphorus (P), sulfur (S), potassium (K), calcium (Ca), manganese (Mn), iron (Fe), zinc (Zn) and iodine (I)] of *Sargassum* spp. is shown in Table 1. The literature shows that seaweed normally contains 10–20-fold more minerals than terrestrial plants since it contains higher concentration rates [24], and this can be used as a complement to other natural fertilizers [25,26]. On the other hand, seaweed has important components for metabolic reactions in human health [27]. The daily intake of approximately 25 g of seaweed replaces the mineral requirement of an adult human being [28].

The presence of heavy metals was observed, among which is arsenic (As), cadmium (Cd), lead (Pb), mercury (Hg), and tin (Sn) (Table 1). According to the results obtained, the seaweed can be a good indicator of the contamination of the surrounding seawater in the region [29], being susceptible to heavy metals due to their strong adsorption properties [30]. The levels of heavy metals detected in the *Sargassum* samples collected are below the levels regulated by the Food Codex STAN 193-1995 and the Official Mexican Standard for food NOM-185-SSA1-2002 (Table 1). According to the above, *Sargassum* could be used as a nutritional source for food and for extraction of beneficial compounds for health.

### 2.3. Hydrothermal Pretreatment of Sargassum spp.

According to the temperatures used for the hydrothermal pretreatment of *Sargassum* spp., a severity factor was obtained for each of the temperatures at 150, 170, and 190 °C of 4.03, 4.13, and 4.20, respectively (Table 2). Figure 1 shows the temperature profiles for high-pressure pretreatments. Table 3 shows the concentration of glucans for each of the temperatures used for hydrothermal pretreatment for 50 min on the *Sargassum* spp. According to the results, the severity factor was higher for the temperature of 190 °C, this is because of the intensity of the process on the *Sargassum* spp. was stronger and allows the glucan to be proportionally greater than when at 150 and 170 °C (Table 3). It was necessary for this study to perform hydrothermal pretreatments; during these processes, polysaccharides are hydrolyzed to monomers [31,32]. This technology reduces operating costs and increases efficiency in the fermentation process; for this reason, it has been explored to optimize saccharification before the bioprocess [33,34]. Figure 2 and Figure 3 show that in the boxplot, it was determined that the growth rate of *Aspergillus* species was higher in the hydrothermal pretreatment of 170 °C for 50 min. The pretreatment of 170 °C on *Sargassum* spp. released 23.86 ± 0.1 g/100 d.w. of glucan, an intermediate amount compared to that obtained at 150 and 190 °C (Table 3). As the severity factor increases, the processing is stronger on the cell wall of the biomass; for this reason, the release of sugars increases (Table 3). Aparicio et al. [14] reported 21.29 ± 0.01, 25.14 ± 0.14, and 32.33 ± 1.20 g/100 g d.w. of glucan in hydrothermal pretreatments of 150, 170, and 190 °C for 50 min, respectively, on *Sargassum* spp. [14]. The results obtained by the author are compared to those obtained in this study. Likewise, the sugars released at a temperature of 170 °C were sufficient to satisfy the requirements for the growth of the fungal strains.

### 2.4. Fungal Radial Growth on Pretreated Sargassum spp. 

The different fungal strains (*Aspergillus oryzae*, *Aspergillus niger HT3*, *Aspergillus niger Aa20* and *Aspergillus niger Aa210*) grew on the pretreated *Sargassum* at 150, 170 and 190 °C, respectively, for 50 min, with (Czapeck) nutrient enrichment (NaNO_3_, MgSO_4_, KCl, and KH_2_PO_4_ g/mL) and without nutritional enrichment. The (Czapeck) nutritional enrichment is a medium for the cultivation of fungi—these compounds are provided by inoculating the *Aspergillus* species so that the inorganic nitrogen is assimilated by the microorganisms for their growth. Consequently, all treatments were effective for the metabolism of filamentous fungi. However, according to the statistical analysis by boxplot, it was inferred that the specific growth rate for *Aspergillus niger Aa20* on *Sargassum* pretreated without enrichment at a temperature of 170 °C was higher, although *Aspergillus oryzae* grew on *Sargassum* pretreated at 170 °C, without nutritional enrichment and with Czapeck. Therefore, *Aspergillus oryzae* was chosen to carry out solid-state fermentation, and the *Sargassum* pretreated at 170 °C as substrate, without enrichment (Figure 2). The results indicate that the hydrothermal pretreatments degraded the structural modifications of polysaccharides on macroalgae cell walls so that they could be assimilated by the microorganisms. According to the above, for solid-state fermentation, hydrothermal pretreatment at 170 °C was chosen without nutritional enrichment. Analysis of variance (ANOVA) was analyzed utilizing MATLAB 2020, where it showed significant differences between temperatures and the growth of the different species of *Aspergillus*. Figure 3 shows the growth of *Aspergillus* on pretreated *Sargassum* with nutritional enrichment at different temperatures (150, 170, and 190 °C, respectively). Figure 4 shows the growth of *Aspergillus oryzae* on pretreated *Sargassum* spp. (150, 170, and 190 °C) without nutritional enrichment.

### 2.5. Solid-State Fermentation on Sargassum spp. Hydrothermally Pretreated at 170 °C

It was observed that the ash content decreased during solid-state fermentation kinetics—15.09% ± 0.2, 14.69% ± 0.2, and 14.02% ± 0.9 at for 24, 72, and 120 h, respectively. The presence of minerals was observed during SSF kinetics (24, 72, and 120 h), the concentration of these elements showed variability during fermentation. Phosphorus (P), sulfur (S), potassium (K), iodine (I), and iron (Fe) decreased in concentration from 24 to 72 h; calcium (Ca) from 24 to 120 h; manganese (Mn) and zinc (Zn) from 72 to 120 h (Table 4). According to Table 4, *Aspergillus oryzae* assimilated heavy metal concentrations from *Sargassum* after hydrothermal pretreatment at 170 °C during SSF; the absorbed elements were As, Cd, Pb, Hg, and Sn. The concentrations of heavy metal are below the limits regulated by the Food Codex STAN 193-1995 and the Official Mexican Standard for food NOM-185-SSA1-2002.

### 2.6. Production of Fungal Proteins from Aspergillus oryzae on Sargassum spp. after Hydrothermal Pretreatment at 170 °C

The highest protein content was found at 96 h (6.6 ± 0.2) (Figure 5) and the lowest at 72 h (4.6 ± 0.4) compared with the control (3.7 ± 0.40). The microorganisms can assimilate the concentration of carbohydrates to form fungal mycelium [35]. This scientific report is consistent with what was found in this study because there was an increase in protein content from 24 h (5.4 ± 0.1%) to 48 h (6.3 ± 0.0). A decrease in the concentration of total sugars was observed from 24 to 120 h (Figure 5). At the beginning of fermentation, the pH was 5.93 ± 0.02, a satisfactory pH for fungal growth according to the literature [36]. According to the analysis of variance (ANOVA), significant differences are observed in the percentage of proteins and total sugars during solid-state fermentation kinetics. Proteins increase due to *Aspergillus oryzae.* These filamentous fungi can depolymerize complex carbohydrates (cellulose, fucoidan, and alginate) [37]. In the same way, these fungi can ferment or grow on the biomass of the seaweed, and assimilate the complex substrates, to produce fungal biomass, rich in proteins and edible—these are called mycoproteins [35].

### 2.7. Production of Fungal Proteins from Aspergillus oryzae on Sargassum spp. after Hydrothermal Pretreatment at 170 °C Using a Packed-Bed Bioreactor

The highest protein content during solid-state fermentation of *Sargassum* spp. using glass columns (packed-bed bioreactor) was at 120 h (8.1 ± 0.12), followed by at 48 h (7.6 ± 0.32). Likewise, the lowest protein content was observed at 72 h (6.4 ± 0.23). Regarding the control, the zero time of SSF kinetics was taken (5.1 ± 0.05) (Figure 6). The protein percentage yield for this fermentation was 3% (5.1–8.1). According to the above, the production of fungal proteins using a packed-bed bioreactor was higher than when using Petri dishes. The results can be attributed to the constant airflow and humidity for the growth of *Aspergillus oryzae* during SSF kinetics.

### 2.8. Scanning Electron Microscopy

The surface morphology of the *Sargassum* was observed using a scanning electron microscope (SEM). A superficial analysis was performed using a semi-quantitative technique. The objective of this analysis was to observe the structural differences that occurred in kinetics times during solid-state fermentation on the pretreated *Sargassum*. Morphological differences are observed in the surface of the *Sargassum* pretreated at 170 °C from the time of growth kinetics of *Aspergillus oryzae* at 24 to 120 h (Figure 7A–D). In the pretreated *Sargassum,* there is evidence of strong coupling in its structure, possibly due to the presence of numerous chemical compositions with various minerals. In Figure 7A–C, small cubes are observed arranged on the surface of the biomass of the seaweed—the residual particles correspond to mineral crystallites. The *Sargassum* spp. has the presence of numerous chemical compositions with various minerals, which originated during its cultivation process. The porous microstructure observed (Figure 7A) may be due to the formation of carbon materials with a lower density [38]. The micrographs of scanning electron microscopy show that the production of proteases from *Aspergillus oryzae* produces a change in the cell wall (Figure 7C,D) [39]. As shown in Figure 7D, at 120 h, the external surface of *Sargassum* spp. thinned and smoothed, because of the metabolic activities of the fungal microorganism. With time, the innermost structure of the cellulose is exposed [40].

## 3. Discussion

The structural polysaccharides of *Sargassum* spp. have great importance for the growth of microorganisms; for this reason, they have been used as a substrate in solid-state fermentation and the production of fungal proteins. Concerning the physicochemical composition of *Sargassum* spp., the results obtained during this study are similar to those reported by Aparicio et al. [14] in a study carried out on *Sargassum* spp. (10.40 g/100 g (d.w.) of glucan, 4.34 of galactan and 6.77 of fucoidan) [14]. Additionally reported is the total carbohydrates for *Himanthalia elongata* (15.69 g/100 g (d.w.)) [41]. Fucoidans include fucose, galactose, xyloid, and mannoses. The composition of fucoidan and galactan in seaweed varies according to species, maturity, location, collection period, and extraction processes [42]. According to the above, among the most abundant components in brown seaweed are carbohydrates—compared to proteins and lipids, important biochemical constituents—since they represent the main source of energy for metabolic routes [27]. The ash content is compared to that reported by Aparicio et al. [14] (20.27 ± 0.44), and those reported by Del Rio et al. [43] (11.87 g/100 g (d.w.)) for *Sargassum* spp. and Sukwong et al. [44] for *Gracilaria verrucosa* (12.43 g/100 g (d.w.)) [14,41,43,44]. The protein content of seaweed can vary by taxa—low (3–15%) in brown seaweed, compared to green and red seaweed. Hence, seaweed contains high ash content compared to edible terrestrial plants, probably associated with the absorption of inorganic compounds and salts in the aquatic environment [45]. The total protein content has been noted for *Sargassum wightii* (8.0 ± 0.17% and 12.2 ± 0.12% d.w.), *Sargassum echinocarpum* (8.7%), *Sargassum henslowianum* (10.3%), *Sargassum mangarevense* (11.3%), *Sargassum obtusifolium* (13%) and *Sargassum thunbergii* (13.9%). Similarly, different species contain high levels of proteins—*Sargassum coreanum* (14.4%), *Sargassum lomentaria* (16.9%), and *Sargassum vulgare* (15.8%). The total protein content of the different vegetative parts gradually decreases from March to September, that is, the total protein content is directly related to the maturation of the thallus [46]. It is worth mentioning that the mineral content of seaweed varies according to the species, geographical origin, season, environmental conditions, and processing method [47]. Studies have reported various concentrations of metals in *Sargassum* such as cadmium (Cd), chromium (Cr), copper (Cu), iron (Fe), lead (Pb), manganese (Mn), and zinc (Zn) [48]. Likewise, recent studies detected elements such as cadmium (Cd), and zinc (Zn) [30] in *Sargassum polycystum*. The absorption of metals by the *Sargassum* can occur through the transfer of free ions to the cell wall, increased transport through the plasma membrane, and diffusion into cells, resulting in an interaction between metals and hydrophilic surfaces [49]. Brown seaweed can bioabsorb and bioaccumulate chemical elements and this is due to the sulfate, amino, carboxyl, and hydroxyl groups present in lipids, proteins, and polysaccharides [50]. Santos et al. [51] mention that the absorption of heavy metals by seaweed is due to the complex cell wall, rich in fucoidanes, alginates, carboxyl groups (up to 80% of its dry weight) [51]. Similarly, heavy metal concentrations in seaweed vary significantly between seaweed species, during the sampling period and site [52].

In the hydrothermal pretreatments, the water penetrates the cell structure, hydrating cellulose, and solubilizing polysaccharides. Monomeric sugars are affected by hydrothermal pretreatment conditions [53,54,55]. In addition, during the operation of this process, other compounds can be produced such as methanol, formic, acetic, propionic, 1-hydroxypropanone, 1 hydroxybutanone, 2 furfuraldehyde and furfural [56,57]. Brown seaweed, compared to other residues, has low bioconversion due to the presence of complex polysaccharides that do not ferment easily, with a mass ratio of carbon and nitrogen below 20:1 and high levels of sulfur, polyphenols, and salinity [58]. Thermal hydrolysis is most advantageous for brown seaweed application and needs to be explored in depth to assess scalability. On the other hand, this promising technique for the treatment of brown seaweed stands out due to its positive aspects that include the increase in methane productivity, the resulting energy balance, generating a biofertilizer, and mitigation of greenhouse gas emissions [34].

The growth of *Aspergillus oryzae* is carried out in a better way through adaptation on solid substrates, since this type of medium is suitable for its development. The above, due to the moisture content, results in the mycelia of the fungus penetrating the *Sargassum*. Similarly, the absence of free water in the solid medium allows this filamentous fungus to acquire the ability to produce metabolites, being hardly produced in a liquid medium. Within the food industry, this microorganism is used for the development of Koji [59]. Now, hydrothermal pretreatment provides selectivity in the separation of the components (soluble polysaccharides can be dissolved without causing significant effects on glucan) [60]. On the other hand, soluble polysaccharides are closely linked to the glucan structure and are located between and within glucan fibrils [61]. Thus, in the hydrothermal pretreatment of the seaweed, carbohydrates remain available, due to the breakdown of polysaccharides present in the cell wall of these, to allow the microorganisms to use these compounds as a source of carbon for their metabolism during solid-state fermentation. During this study, it was found that the different temperatures (150, 170, and 190 °C) used in the hydrothermal pretreatments were effective for the growth of *Aspergillus*, but there are more suitable conditions for the development of these fungi, as was the case with pretreatment at 170 °C. Pérez-Larrán et al. [62] reported that the saccharide content reaches 33% at this temperature, and the maximum fucose content, representing more than 30% of the total sugar content, was observed at 150 °C [62]. The literature reports the presence of 2-furfural after hydrothermal treatments, due to the chemical and thermal conversion of C5 sugars [63], so that certain temperatures can inhibit cell growth, as a consequence of the conversion of glucose released by cellulose into hydromethylfurfural [64].

The literature states that fermentation is an efficient route to desorb heavy metals and convert complex sugars into simple sugars, to be easily assimilated by microorganisms to obtain a higher digestion rate [65]. Additionally, microorganisms consume nutrients to grow and produce more cells and important products. These can grow under different physical factors, chemical, and nutritional conditions. In a suitable nutrient medium, organisms extract nutrients from the medium and convert them into biological compounds [36]. The biosorption and bioaccumulation of metal ions by living or dead microbial cells have been described in different studies [66]. Cd, Cu, Cr, Zn, Pb, and Ni were eliminated from wastewater by native microorganisms [67]. The decrease in the concentration of heavy metals, such as lead (Pb) at 13.84% and cadmium (Cd) at 31.08%, results from the effect of *Phanerochaete chrysosporium* on the co-composting of agricultural waste and river sediments [68].

With regard to the aforementioned, it is possible to interpret that *Aspergillus oryzae* fulfills important functions in the bioconversion of the biomass of *Sargassum* spp. through solid-state fermentation, since the concentration of heavy metals present in *Sargassum* spp. decreases due to the metabolism of these microorganisms. It is necessary to consider that before solid-state fermentation, the levels of heavy metals were below the limits allowed by the Food Codex and the Official Mexican Standard. As such, the biotechnological process developed (solid-state fermentation) allows changes to be made on the levels of metallic ions to increase the added value of *Sargassum* spp. in the food industry.

Normally, fungi can produce extracellular enzymes to hydrolyze the complex polymers of brown seaweed and release carbon, nitrogen, and other nutrients for growth and metabolic functions [35,69,70]. The protein content has been attributed to the enzymatic reactions to degrade the polymers to mono or short oligomers and assimilate them, including the nitrogen compounds of the substrate, and thus promote alterations in solubility and nitrogen concentrations [37,71]. This metabolic process is also due to carbohydrates that are degraded and converted into fungal biomass and CO_2_, which leads to an increase in nitrogen content [35,71]. The production of fungal proteins from biomass represents the decrease in carbohydrate content during growth, regardless of the fungus used [70,72]. *Sargassum* spp. is currently considered an organism with various effects on the environment; for this reason, the fungal protein produced from this seaweed could be a protein alternative or protein ingredient for food. *Aspergillus oryzae* produces amylases and proteases during the different phases of growth. According to studies, amylases are excreted at the beginning of the metabolic process (0–18 h), since they are enzymes necessary for the microorganism to obtain carbohydrates (sugars) for growth. Similarly, proteases are produced in a second phase (18–48 h), that is, when the microorganism has grown competently and has consumed carbohydrates available in the solid medium [73]. The pH plays an important role in the growth and metabolism of the microorganism [74]. However, during the investigation, it was shown that the pH of the pretreated *Sargassum* spp. positively influenced the process, because the growth of microorganisms was optimal.

Solid-state fermentation produces nutritional improvements, among which is the enrichment of crude protein and organic acids as well as degradation of tannins, dietary fiber, and phytic acid [75]. The literature reports that microorganisms consume sugars and hydrolyze the long chains of carbohydrates in waste to produce biomass rich in protein [23]. According to studies, the composition of the biomass of the mycelium is not constant. The polysaccharide and protein content metabolized by the fungus can vary considerably. This is related to genetic affiliation, the productivity of the strains used, culture conditions, and media composition [76]. Oluwaseye et al. [75] reported an increase in protein (1.77% and 0.3%) using *Aspergillus sojae* and *Aspergillus ficuum*, respectively, in solid-state fermentation of canola meal [75]. Under optimized solid-state fermentation conditions using sago waste, a 1.3% increase in crude protein was obtained [77]. The protein content of pineapple peels using *Trichoderma viride* was 16% when using bioreactors with aeration (packed-bed bioreactor), and 14.89% when using a conical flask [78]. Ezekiel et al. [79] reported higher amounts of cassava peel protein fermented with *Trichoderma viride* using a farm-scale bioreactor [79]. These results are due to adequate aeration by the air pump and the relative humidity that was maintained during fermentation in the bioreactor [78]. Brain et al. [80] produced microbial biomass rich in protein using the green macroalga *Ulva* rigida as a substrate through submerged fermentation and using *Trichoderma reesei* for such a process. They showed the consumption of glucose by *Trichoderma reesei* after 24 h of culture and an increase in fungal biomass and proteins; in the same way, certain microorganisms decrease the pH of carbohydrate substrates when inoculated [80]. These results are similar to those obtained in this investigation, since a decrease in sugars and an increase in protein were observed at 24 h (Figure 5). In this study, *Aspergillus oryzae* was inoculated on the *Sargassum* substrate hydrothermally pretreated at 170 °C for 50 min, resulting in a pH of 5.93 ± 0.02. Solid-state fermentation provides different positive aspects to biomass, since it increases yield and productivity, and improves product characteristics through bioactive compounds. This happens because the microorganisms modify the constituents of the substrate through fermentation. During this bioprocess, many biochemical changes occur, affecting the bioactive and digestibility of the product [81]. Microorganisms play essential roles in the conversion of valuable products from macroalgae; for this reason, metabolic engineering technologies are in development. The sugar content affects the physiology and biochemistry of microorganisms because sugar catabolism involves the coordinated expression of diverse pathways and extensive metabolic engineering needs [82]. A study was conducted using *Bacillus* species in soybean meal to produce bacterial protein through solid-state fermentation, resulting in a yield of 442.4 to 524.8, 516.1 and 499.9 g kg^−1^, and from 53.9 to 203.3, 291.3 and 74.6 g kg^−1^ [83]. When developing the investigation, the presence of 23.86 ± 0.1 of glucan was determined in the pretreatment of 170 °C for 50 min, polysaccharides provided carbon sources for the growth of *Aspergillus oryzae*, and the microorganism also carried out metabolic processes for the conversion of the fungal protein. During solid-state fermentation kinetics, a total sugar concentration of 25.82 ± 0.007 was determined at 0 h and, as time passed, the sugar content decreased. Fungi can use various polymeric carbohydrates, and these microorganisms breakdown different carbohydrate molecules in aerobic and anaerobic environments during fermentative processes [84]. For decades, seaweed have been used as protein sources for different industries [85].

Scanning electron micrographs reveal relevant characteristics in the structure and morphology of *Sargassum* spp. through solid-state fermentation kinetics. In solid-state fermentation, one of the main obstacles to cellulose degradation is the structural properties of the biomass, but the destruction of the cell wall structure, alteration of the morphology, and formation of cells were observed through SEM, showing holes in the biomass surfaces [86]. Microbial fermentation produces structural modifications—the holes and cracks observed after SSF could be due to the degradation of polysaccharides in the cell wall and the structure of the biomass [87]. The structure of polysaccharides in seaweed differs from polysaccharides in terrestrial plants since seaweed has strong anionic sulfate groups, which exert a certain effect on the morphological structure [88]. In the pretreated *Sargassum*, morphological changes are observed on the surface of the seaweed, showing significant differences between the cellulose before and after solid-state fermentation [89]. Figure 7 shows that *Sargassum* spp. after hydrothermal pretreatment at 170 °C had a more compact and smoother fibrillar structure, while seaweed subjected to solid-state fermentation showed changes in the morphology of the cell wall. The fibrous cap was degraded, possibly due to the decreased crystallinity of the cellulose. The changes indicate that enzymes metabolized by *Aspergillus oryzae* broke down the cell wall [90]. The difference in the morphology of the substrate through solid-state fermentation kinetics may occur due to cellular physiology in this bioprocess, with hydrophobic conidia playing a very important role in the viability and dispersion of fungi [91].

## 4. Materials and Methods

### 4.1. Sargassum Biomass Collection and Preparation

The *Sargassum* spp. (*S.* spp.) utilized in this work was collected in Puerto Morelos, Quintana Roo, México (GPS Coordinates: 20.83149359, 86.87929434) in February 2019. The *S.* spp. was washed with tap water, dried in the sun for 48 h, and then taken to a mill (Thomas Wiley, Swedesroro, NJ, USA). The samples were then sieved, using a 2 mm screen, retaining particles between 300 and 500 µm. These samples were stored in Ziploc bags. To determine the moisture content, 1 g of *S.* spp. was used at 55–60 °C for 24 h to measyre the difference between dry and wet weight [92].

### 4.2. Physicochemical Characterization of Sargassum biomass

The biochemical characterization of the biomass of *Sargassum* spp. was determined according to crude protein and ash content [92,93]. Total sugars were determined using Antrona through the methodology of [94].

The polysaccharide (glucan, galactan, and fucoidan) content was determined by quantitative acid hydrolysis at 72% (*w*/*v*), following the standard analytical procedure of the National Renewable Energy Laboratory [95]. An amount of 0.5 g of dry biomass was added to test tubes with 5 mL of 72% sulfuric acid (*v*/*v*) and then placed in a water bath at 35 °C with manual stirring for one hour. Subsequently, the samples were diluted to 4% sulfuric acid (adding 148.67 g of distilled water) and placed in an autoclave for 1 h. The samples were brought to room temperature before filtering a 1 mL aliquot with a 0.45 μm nylon filter into a vial to determine the release of monomeric sugars by HPLC. The of polysaccharide content in the samples was determined: glucose as laminarin (as glucan), galactose as galactan, and fucose as fucoidan. Finally, the solid fraction was dried in an oven at 50 °C for 24 h to determine the acid-insoluble residue [43].

High-performance liquid chromatography (HPLC) analysis was performed in an Agilent 1260 Infinity II with a refractive index for glucose, fucose, and galactose using the calibration curves of these pure compounds to determine their concentrations. A MetaCarb 87 H column (300 mm × 7.8 mm, Agilent, Ratingen, Germany) was used for the analysis with a column temperature of 60 °C and 0.005 mol/L sulfuric acid as the mobile phase, at a flow rate of 0.7 mL/min.

#### 4.2.1. Elemental Composition Characterization

The mineral composition of the *Sargassum* spp. was estimated by X-ray fluorescence (Equipment Panalytical, Epsilon 1, Almelo (The Netherlands), using a spectrometer and the Omniam Software.

#### 4.2.2. pH

An amount of 1 g of ground pelagic *Sargassum* spp. was placed in 10 mL of distilled water for 30 min. After this time, the pH was determined using a potentiometer [92].

### 4.3. Hydrothermal Pretreatment of Sargassum spp. Biomass

Hydrothermal pretreatment was performed in a pressurized stainless steel batch reactor, equipped with a stirrer and a proportional-integral-derived temperature controller (PID). The reactor is equipped with a temperature sensor (thermocouple with thermowell) and a pressure sensor (dry manometer) with a volume of 0.662 L. This reactor is a conceptual, basic, and detailed engineering design, developed by the biorefinery group (www.biorefinerygroup.com; accessed on 15 February 2022). The reactor was heated with an electrical resistance and a water jacket cooling system was used to lower the temperature of the reactor. *Sargassum* and water were mixed at a ratio of 1:10 (*w*/*v*) and placed in the reactor, heated, and held for the residence time; the sample was stirred at 200 rpm; after that, the reactor cooled down. The high-pressure hydrothermal pretreatments were carried out under different operating temperature (150, 170, and 190 °C), pressure (3.75–11.54 bar), and residence times (50 min). The hydrothermal pretreatments were carried out in triplicate. After the reactor cooled down, the solid and liquid phases were separated by filtration. The solid phase was washed with distilled water was used to remove any degradation compounds that were present, and then the samples were dried in a cold-air oven. The solid phase that remained after hydrothermal pretreatment was then used to calculate the solid yield (g of solid recovered/100 g of initial *Sargassum* spp.). Then, the polysaccharide content was analyzed through the quantitative acid hydrolysis methodology. The intensity of hydrothermal pretreatment was expressed by severity factor, considering different operating temperatures and times—this factor can be expressed by the following equations [53,54]:(1)$$LogR_{o}=\left[R_{o}Heating\right]+\left[R_{o}Isothermal\;process\right]+\left[R_{o}Cooling\right]$$
(2)$$LogR_{o}=\left[\overset{tMAX}{\underset{0}{\int }}\frac{T\left(t\right)-100}{\omega }dt\right]+\left[\overset{ctrf}{\underset{ctri}{\int }}exp\left[\frac{T\left(t\right)-100}{\omega }\right]dt\right]+\left[\overset{0}{\underset{ctrf}{\int }}\frac{T\left(t\right)-100}{\omega }dt\right]$$

### 4.4. Preparation of the Inoculum and Radial Growth of Aspergillus fungi

Four species of the genus *Aspergillus* (*niger HT3*, *niger Aa20*, *niger Aa210*, and *oryzae*) were obtained from the mycotic of the Autonomous University of Coahuila; these were grown on potato dextrose agar (PDA) in 250 mL Erlenmeyer flasks for 7 days at a temperature of 30 °C, an inoculum size of 1 × 10^6^ spores/g and a humidity of 70%. *Aspergillus* were inoculated on 3 g of pretreated biomass (150, 170, and 190 °C) for 50 min, observing growth kinetics and taking measurements (radial) every 12 h for 5 days. The methodology described by Londoño et al. [74] was used to obtain the spores. The spores were counted in the Neubauer cell [74]. Likewise, the biomass after hydrothermal pretreatment without nutritional enrichment and *Sargassum* spp. pretreated and enriched with Czapeck (NaNO_3_, MgSO_4_, KCl, and KH_2_PO_4_ g/mL) were used. In accordance with the above, to continue solid-state fermentation and identify the most effective hydrothermally pretreatment in the growth of *Aspergillus*, a boxplot was constructed using the MATLAB 2020 program.

### 4.5. Solid-State Fermentation on Pretreated Sargassum spp. Biomass

A spore concentration of 1 × 10^6^ spores/g of *Aspergillus oryzae* was inoculated on the hydrothermally pretreated *Sargassum* spp. at 170 °C without nutritional enrichment, at a temperature of 30 °C and moisture of 70%. The experiments were carried out using Petri dishes, where 6 g of dry pretreated biomass was taken and moistened with water at the indicated value. The samples were inoculated with prepared spores and subsequently incubated at 30 °C for 5 days. Samples were taken every 24 h to evaluate the fungal protein and total sugar content in fermentation kinetics. Figure 8 shows the schematic representation of the hydrothermal process and solid-state fermentation on pretreated *Sargassum* spp. biomass.

### 4.6. Solid-State Fermentation (Packed-Bed Bioreactor) in Pretreated Sargassum spp. Biomass

For solid-state fermentation of *Sargassum* spp. pretreated at 170 °C, glass columns with a diameter of 2.5 mm were used. Within each of the columns, 10 g of pretreated *Sargassum* spp. was added. Likewise, the system had a constant temperature of 30 °C and a humidity of 70%. In the same way, moist air was added to the columns using humidifiers connected to an air pump (Figure 9).

### 4.7. Scanning Electron Microscopy in the Fermented Biomass of Sargassum spp.

The surface morphology of the samples was characterized using an environmental scanning electron microscope (Philips XL30-ESEM—Mode GSE Detector [Gaseous Secondary Electrons). To evaluate the surface morphology, samples were mounted in sample holders, and then the samples adhered to the sample holder with double-sided carbon tape so that the samples were positioned perpendicular to the microscope detectors.

### 4.8. Statistical Analyses

The experiments were performed in three repetitions, and the results were described as the mean ± standard deviation of the mean (SD). Significant differences (*p* ≤ 0.05) between values were subjected to a one-way analysis of variance and tested using the Tukey test. Analysis of variance (ANOVA) was performed using MATLAB 2020.

## 5. Conclusions

This study demonstrated the potential to produce fungal proteins through solid-state fermentation using *Aspergillus oryzae* as the substrate of *Sargassum* spp. after hydrothermal pretreatment at 170 °C for 50 min. By using hydrothermal pretreatment, the biomass of *Sargassum* spp. is fractionated, thus leaving glucans available to be assimilated by microorganisms during solid-state fermentation. It is necessary to vary pH, inoculum size, particle size, humidity, and temperature to optimize performance to produce fungal proteins. Through scanning electron micrographs of *Sargassum* spp., fermented morphological changes were observed due to the efficient growth of *Aspergillus oryzae* on the substrate. According to the above, a combination of biotechnological processes (hydrothermal pretreatment and solid-state fermentation) is effective to give added value to *Sargassum* spp. To sum, *Sargassum* spp. is an efficient substrate to produce fungal proteins for the food industry.

## Figures and Tables

**Figure 1 molecules-27-03887-f001:**
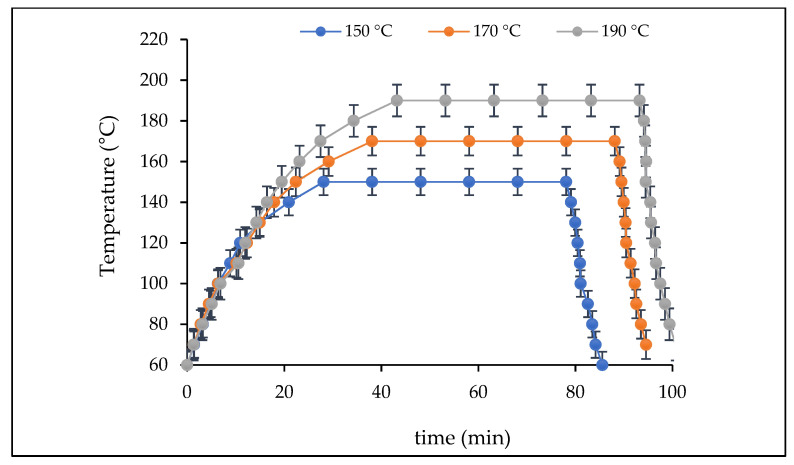
Temperature change profile during hydrothermal pretreatment (150, 170 and 190 °C, respectively, for 50 min).

**Figure 2 molecules-27-03887-f002:**
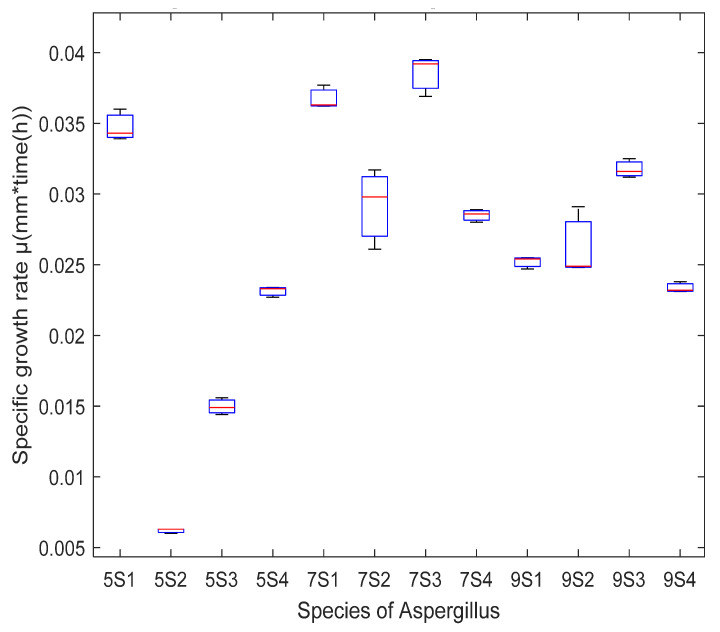
Boxplot of the specific growth rate of *Aspergillus* species on *Sargassum* spp. after hydrothermal pretreatment at 150, 170, and 190 °C without nutritional enrichment (5S1: 150 °C, *Aspergillus oryzae*; 5S2:150 °C, *Aspergillus niger HT3*; 5S3: 150 °C, *Aspergillus niger Aa20*; 5S4: 150 °C, *Aspergillus niger Aa210*; 7S1: 170 °C, *Aspergillus oryzae*; 7S2:170 °C, *Aspergillus niger HT3*; 7S3: 170 °C, *Aspergillus niger Aa20*; 7S4: 170 °C, *Aspergillus niger Aa210*; 9S1: 190 °C, *Aspergillus oryzae*; 9S2:190 °C, *Aspergillus niger HT3*; 9S3: 190 °C, *Aspergillus niger Aa20*; 9S4: 190 °C, *Aspergillus niger Aa210*).

**Figure 3 molecules-27-03887-f003:**
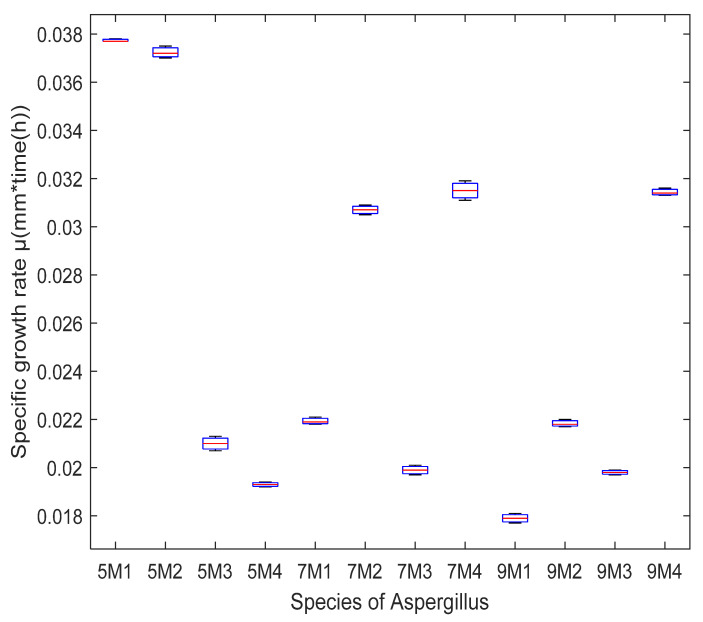
Boxplot of the specific growth rate of *Aspergillus* species on *Sargassum* spp. after hydrothermal pretreatment at 150, 170, and 190 °C with nutritional enrichment (5M1: 150 °C, *Aspergillus oryzae*; 5M2:150 °C, *Aspergillus niger HT3*; 5M3: 150 °C, *Aspergillus niger Aa20*; 5M4: 150 °C, *Aspergillus niger Aa210*; 7M1: 170 °C, *Aspergillus oryzae*; 7SM2:170 °C, *Aspergillus niger HT3*; 7M3: 170 °C, *Aspergillus niger Aa20*; 7M4: 170 °C, *Aspergillus niger Aa210*; 9M1: 190 °C, *Aspergillus oryzae*; 9M2:190 °C, *Aspergillus niger HT3*; 9M3: 190 °C, *Aspergillus niger Aa20*; 9M4: 190 °C, *Aspergillus niger Aa210*).

**Figure 4 molecules-27-03887-f004:**
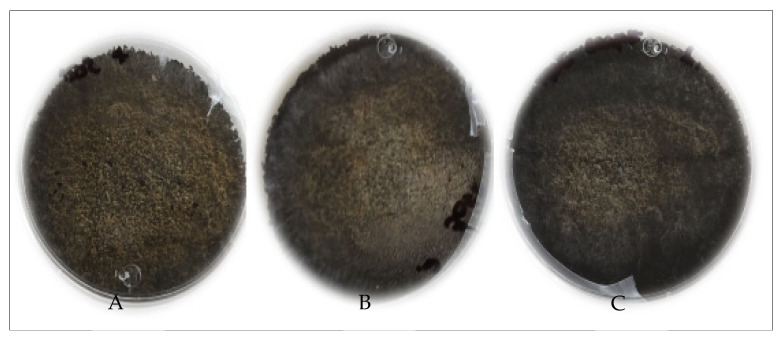
Growth of *Aspergillus oryzae* on *Sargassum* spp. after hydrothermal pretreatment at 150 (**A**), 170 (**B**) and 190 °C (**C**) without nutritional enrichment.

**Figure 5 molecules-27-03887-f005:**
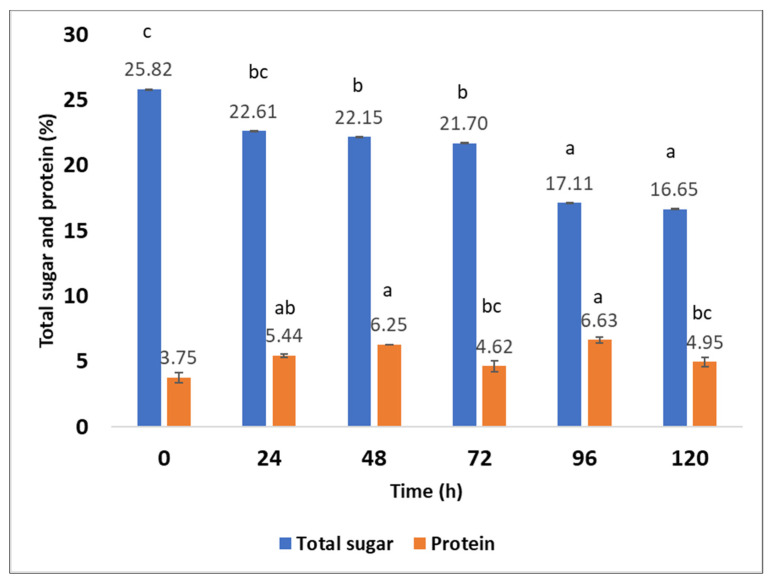
Fungal proteins and total sugars obtained from solid-state fermentation of *Sargassum* spp. after hydrothermal pretreatment (170 °C). The treatments that do not have the same letters (a, b, and c) are significantly different.

**Figure 6 molecules-27-03887-f006:**
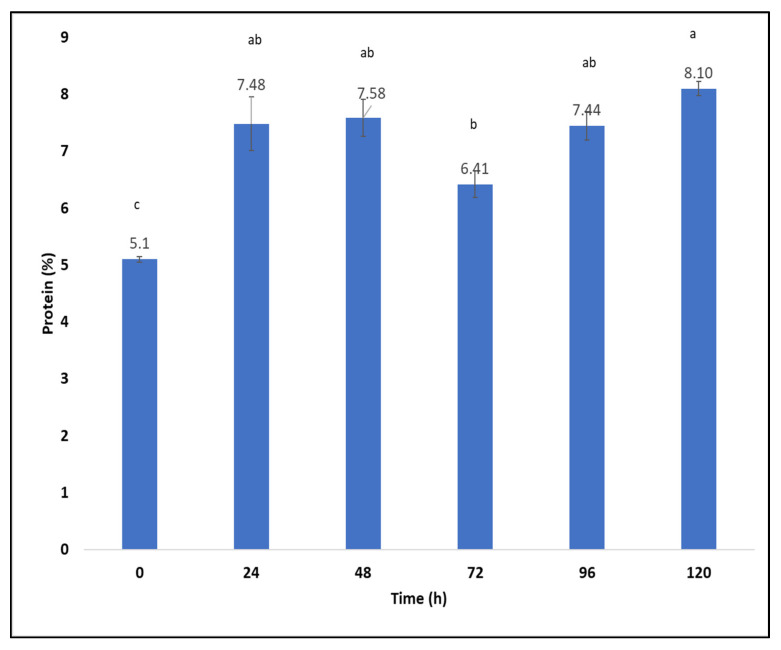
Fungal proteins obtained during SSF kinetics using *Sargassum* spp. after hydrothermal pretreatment (170 °C) as substrate in a packed-bed bioreactor. The treatments that do not have the same letters (a, b, and c) are significantly different.

**Figure 7 molecules-27-03887-f007:**
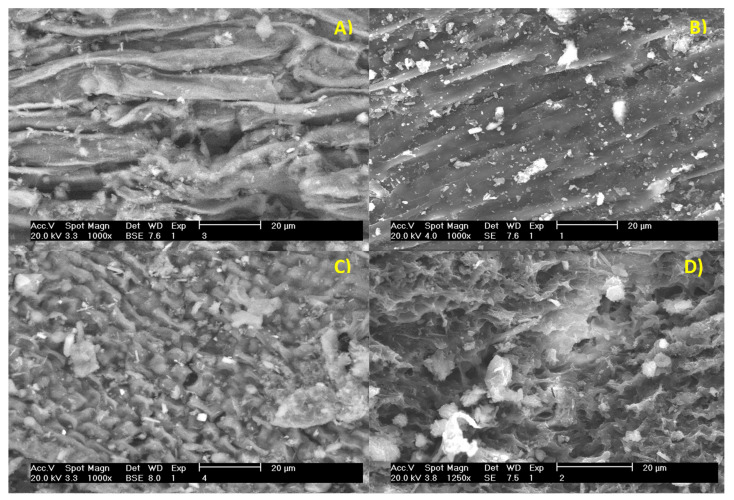
Scanning electron microscopy (SEM) (**A**) of *Sargassum* after hydrothermal pretreatment at 170 °C; (**B**) *Aspergillus oryzae* growth kinetics at 24 h through solid-state fermentation; (**C**) *Aspergillus oryzae* growth kinetics at 72 h through solid-state fermentation; (**D**) *Aspergillus oryzae* growth kinetics at 120 h through solid-state fermentation.

**Figure 8 molecules-27-03887-f008:**
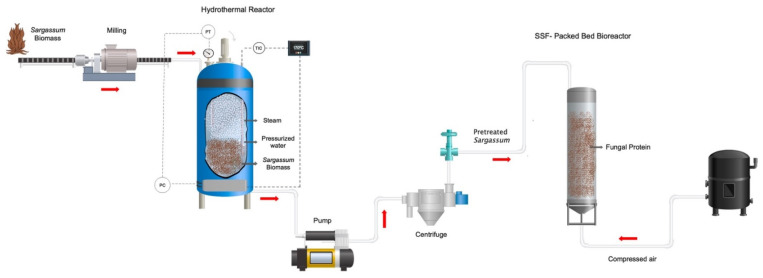
Schematic representation of the general process in the production of fungal proteins using a sequential process: hydrothermal pretreatment and solid-state fermentation.

**Figure 9 molecules-27-03887-f009:**
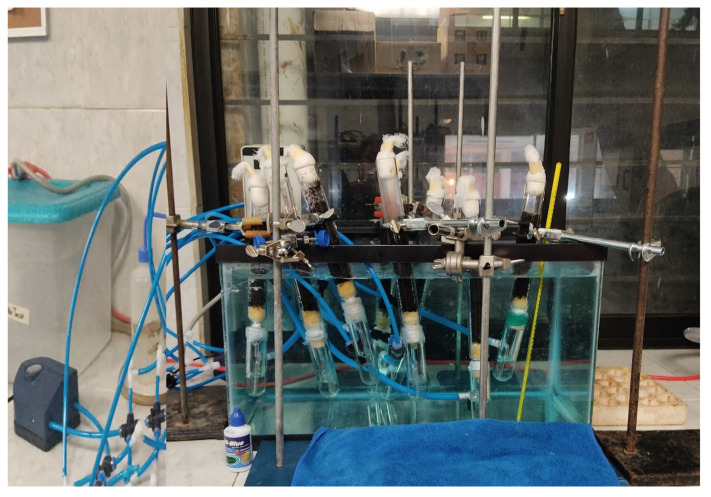
Solid-state fermentation system (packed-bed bioreactor) for *Sargassum* spp. pretreated at 170 °C to obtain fungal proteins from *Aspergillus oryzae*.

**Table 1 molecules-27-03887-t001:** Physicochemical and elemental characterization of *Sargassum* biomass.

Chemical Composition	Seaweed g/100 g (d.w.)	Minerals	Concentration (mg/kg)	Heavy Metals	Concentration (mg/kg)	Codex Alimentarius (mg/kg)	Official Mexican Standard for Food (mg/kg)
Protein (%)	4.3 ± 0.07	Magnesium (Mg)	13.45 ± 0.00	Arsenic (As)	8.81 × 10^−2^ ± 0.00	0.2	0.2
Glucan	11.64 ± 0.01	Phosphorus (P)	2.44 ± 0.00	Cadmium (Cd)	1.82 × 10^−3^ ± 0.00	0.4	
Galactan	0.99 ± 0.01	Sulfur (S)	15.51 ± 0.00	Lead (Pb)	2.32 × 10^−5^ ± 0.00	0.2	0.1
Fucoidan	3.18 ± 0.01	Potassium (K)	3.12 ± 0.00	Mercury (Hg)	4.10 × 10^−3^ ± 0.00	0.1	0.05
Insoluble acid residue	29 ± 0.01	Calcium (Ca)	136.16 ± 0.00	Tin (Sn)	3.25 × 10^−2^ ± 0.00	250; 150; 50	250
Ash	15.46 ± 0.40	Manganese (Mn)	0.11 ± 0.00				
		Iron (Fe)	0.31 ± 0.00				
		Zinc (Zn)	0.11 ± 0.00				
		Iodine (I)	0.17 ± 0.00				

**Table 2 molecules-27-03887-t002:** Operating conditions for hydrothermal processing of *Sargassum* spp.

Operating Conditions	150 °C	170 °C	190 °C
Time (min)	50	50	50
Pressure (bar)	3.75	6.91	11.54
Severity process [log*R*_o_]	4.03	4.13	4.20

**Table 3 molecules-27-03887-t003:** Chemical characterization of solid phase (g/100 g d.w.) for hydrothermal pretreatment of *Sargassum* spp.

		Temperature (°C)	
Chemical components	150	170	190
Glucan	17.99 ± 0.1	23.86 ± 0.1	25.38 ± 0.1

**Table 4 molecules-27-03887-t004:** Concentration of minerals and heavy metals present during solid-state fermentation of *Sargassum* spp.

**Minerals**	**Time (h)**		
**(mg/kg)**	**24**	**72**	**120**
Phosphorus (P)	1.71 × 10^−4^ ± 0.00	1.34 × 10^−4^ ± 0.00	1.51 × 10^−4^ ± 0.10
Sulfur (S)	4.33 × 10^−4^ ± 0.00	3.73 × 10^−4^ ± 0.00	3.86 × 10^−4^ ± 0.00
Potassium (K)	1.36 × 10^−5^ ± 0.00	1.18 × 10^−5^ ± 0.00	2.59 × 10^−5^ ± 0.00
Calcium (Ca)	1.33 × 10^−2^ ± 0.00	1.30 × 10^−2^ ± 0.00	1.22 × 10^−2^ ± 0.00
Manganese (Mn)	1.58 × 10^−5^ ± 0.00	1.76 × 10^−5^ ± 0.00	1.47 × 10^−5^ ± 0.00
Iron (Fe)	7.32 × 10^−5^ ± 0.00	5.36 × 10^−5^ ± 0.00	1.05 × 10^−4^ ± 0.00
Zinc (Zn)	9.81 × 10^−6^ ± 0.00	1.25 × 10^−5^ ± 0.00	1.12 × 10^−5^ ± 0.00
Iodine (I)	3.77 × 10^−6^ ± 0.00	2.20 × 10^−6^ ± 0.00	4.91 × 10^−6^ ± 0.00
**Heavy Metals**	**Time (h)**		
**(mg/kg)**	**24**	**72**	**120**
Arsenic (As)	3.24 × 10^−6^ ± 0.00	3.23 × 10^−6^ ± 0.00	1.15 × 10^−6^ ± 0.00
Cadmium (Cd)	2.42 × 10^−3^ ± 0.00	4.40 × 10^−5^ ± 0.00	2.10 × 10^−3^ ± 0.00
Lead (Pb)	5.09 × 10^−3^ ± 0.00	4.50 × 10^−3^ ± 0.00	3.10 × 10^−3^ ± 0.00
Mercury (Hg)	1.40 × 10^−3^ ± 0.00	2.90 × 10^−3^ ± 0.00	2.68 × 10^−3^ ± 0.00
Tin (Sn)	1.81 × 10^−6^ ± 0.00	1.91 × 10^−6^ ± 0.00	1.61 × 10^−6^ ± 0.00

## Data Availability

Not applicable.

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
