# Peer review of "Fungal Proteins from Sargassum spp. Using Solid-State Fermentation as a Green Bioprocess Strategy"

_molecules, 2022, doi:10.3390/molecules27123887_

Round 1

Reviewer 1 Report

This research studied the utilization of Sargassum app with SSF to obtain fungi proteins.  Overall, author provide some valid results, but the discussion and the explanation in this paper requires more work.

In this study, the materials were pretreated with 3 different temperature. In section 2.3, author conclude that the 170 is the optimal pretreatment condition.  The results and discussion didn't support the statement.  Line 118 - 123 is difficult for reader to understand.

In section 2.4, the SSF study was performed with material pretreated under 3 temperatures.  The explanation of the results are not sufficient here. 

In section 2.5, why the standard deviation for mineral contents in table 6&7 are all zeros? Were there any replicates performed for SSF?

Author Response

Decision Letter - Molecules-1753317

Dear Editor

Molecules Journal

We are sending you the revised manuscript for the paper entitled “Fungal Proteins from Sargassum spp. Using Solid-State Fermentation as a Green Bioprocess Strategy”. We would like to thank the reviewers for the comments and suggestions given to improve the quality of our manuscript. We think that the suggested modifications were appropriate and therefore they were included in the text. Also, they helped us in improving the text in other sections. We believe that this new version of the manuscript is much better than that previously submitted. Our responses for each one of the reviewers’ comments are given below (point by point). The changes made in the new version of the manuscript are highlight in yellow.

ANSWERS TO THE REVIEWERS

 Reviewer # 1:

This research studied the utilization of Sargassum app with SSF to obtain fungi proteins.  Overall, author provide some valid results, but the discussion and the explanation in this paper requires more work.

We agree with the reviewer’s suggestion. We have added more discussion in this section.

In this study, the materials were pretreated with 3 different temperatures. In section 2.3, author conclude that the 170 is the optimal pretreatment condition.  The results and discussion didn't support the statement.  Line 118 - 123 is difficult for reader to understand.

We thank the reviewer’s comment. We have made modifications for a better understanding.

In section 2.4, the SSF study was performed with material pretreated under 3 temperatures.  The explanation of the results are not sufficient here.

We thank the reviewer’s comment. This section has been modified.

In section 2.5, why the standard deviation for mineral contents in table 6&7 are all zeros? Were there any replicates performed for SSF?

We appreciate the reviewer's comments, the deviations are very extremely small, so putting 2 decimal places values is zero. The measurements were made in triplicate, the equipment where the measurements were made is quite precise.

Reviewer 2 Report

Molecules Journal

Fungal proteins from Sargassum spp. using solid-state fermentation as a green bioprocess strategy

Title: uppercase the first letter of each word

Abstract:

Lines 13, and 14 rewrite for the clarity

Lines 13-16, brief and add relevant results instead of that

Line 18, that present

Line 19, the results indicated that a pretreatment temperature, of 170°C was suitable for fungal growth

Line 20, SSF technique

There are many language mistakes, check all over the  manuscript for linguistic errors

Introduction:

Line 29, Sargassum spp

Line 30, Delete “being found”

Line 32 moves citations 2,3 at the end of the sentence next to citation 4

Line 33, clear “eutrophication in marine ecosystems around the world an environmental problem”

Line 51, g/100g, and delete space after the citation

The introduction is inclusive but the language is deplorable, so linguistically revised the manuscript by a native person

Results:

Line 81, because of the unique composition of Sargassum

I suggest merging the first three tables into one table as the chemical composition of sargassum

This study needs to add the analysis of carbohydrates in sargassum before the fermentation as you mention that those carbohydrates are essential for fermentation or add previous studies about the sugars in sargassum

Line 133, makes clear that is an elemental enrichment media

Enhance figure 4

Merge tables 6,7

Reformulate figure 5 also figure 6

Update citations in discussion up to 2022

Line 430, One gram of ..

Add the origin of all devices in manuscript

References don’t follow the journal structure, reformulate

Check the outputs of all references

Author Response

Decision Letter - Molecules-1753317

Dear Editor

Molecules Journal

We are sending you the revised manuscript for the paper entitled “Fungal Proteins from Sargassum spp. Using Solid-State Fermentation as a Green Bioprocess Strategy”. We would like to thank the reviewers for the comments and suggestions given to improve the quality of our manuscript. We think that the suggested modifications were appropriate and therefore they were included in the text. Also, they helped us in improving the text in other sections. We believe that this new version of the manuscript is much better than that previously submitted. Our responses for each one of the reviewers’ comments are given below (point by point). The changes made in the new version of the manuscript are highlight in yellow.

ANSWERS TO THE REVIEWERS

Reviewer # 2:

Title: uppercase the first letter of each word

We thank the reviewer’s comment. We have made modifications in the title.

Abstract:

Lines 13, and 14 rewrite for the clarity

We thank the reviewer’s comment. We have made modifications in this section.

Lines 13-16, brief and add relevant results instead of that

We thank the reviewer’s comment. We have made modifications in this section.

Line 18, that present

We thank the reviewer’s comment. We have made modifications in this section.

Line 19, the results indicated that a pretreatment temperature, of 170°C was suitable for fungal growth

We thank the reviewer’s comment. We have made modifications in this section.

Line 20, SSF technique

We thank the reviewer’s comment. We have made modifications in this section.

There are many language mistakes, check all over the manuscript for linguistic errors

We thank the reviewer’s comment. In grammatical terms, the MS has been revised and modified for better understanding. Thank you.

Introduction:

Line 29, Sargassum spp

We thank the reviewer’s comment. We have made modifications in this section.

Line 30, Delete “being found”

We thank the reviewer’s comment. We have made modifications in this section.

Line 32 moves citations 2,3 at the end of the sentence next to citation 4

We thank the reviewer’s comment. We have made modifications in this section.

Line 33, clear “eutrophication in marine ecosystems around the world an environmental problem”

We thank the reviewer’s comment. We have made modifications in this section.

Line 51, g/100g, and delete space after the citation

We thank the reviewer’s comment. We have made modifications in this section.

The introduction is inclusive but the language is deplorable, so linguistically revised the manuscript by a native person

We thank the reviewer’s comment. In grammatical terms, the MS has been revised and modified for better understanding. Thank you.

Results:

Line 81, because of the unique composition of Sargassum

We thank the reviewer’s comment. We have made modifications in this section.

I suggest merging the first three tables into one table as the chemical composition of sargassum

We thank the reviewer’s comment. We have made modifications in this table.

This study needs to add the analysis of carbohydrates in sargassum before the fermentation as you mention that those carbohydrates are essential for fermentation or add previous studies about the sugars in sargassum.

We appreciate the reviewer's comment. We determined the glucan content before and after the hydrothermal treatment with the intention of showing the effect on the increase of this glucose chain. In addition to this, the start of growth kinetics of Aspergillus oryzae, Aspergillus niger HT3, Aspergillus niger Aa20 and Aspergillus niger Aa210 on the pretreated Sargassum was performed to determine the best adaptation (on the carbon-glucan source) of the microorganism, in this section a statistical analysis was performed.

Line 133, makes clear that is an elemental enrichment media

We thank the reviewer’s comment. We have made modifications in this section.

Enhance figure 4

Thank you very much for your comments, the figure was modified

Merge tables 6,7

We agree with the reviewer, the table 6 and 7 were placed in the same table

Reformulate figure 5 also figure 6

We agree with the reviewer, the figures were edited

Update citations in discussion up to 2022

References have been updated.

Line 430, One gram of ..

This line has been removed

Add the origin of all devices in manuscript

Information regarding some equipment has been added

References don’t follow the journal structure, reformulate

We have revised the edition of the references, thank you very much

Check the outputs of all references

We have revised the references, thank you very much

Round 2

Reviewer 1 Report

the changes looks good

Reviewer 2 Report

Dear Authors

Thanks for responding to the suggested correction